# Association of Mutations in Replicative DNA Polymerase Genes with Human Disease: Possible Application of *Drosophila* Models for Studies

**DOI:** 10.3390/ijms24098078

**Published:** 2023-04-29

**Authors:** Masamitsu Yamaguchi, Sue Cotterill

**Affiliations:** 1Technical Division, Kankyo Eisei Yakuhin Co., Ltd., Kyoto 619-0237, Japan; myamaguc8@gmail.com; 2Molecular and Clinical Sciences Research Institute, St George’s University of London, London SW17 0RE, UK

**Keywords:** replicative DNA polymerase genes, DNA polymerase α-primase, DNA polymerase δ, DNA polymerase ε, *Drosophila* model, human diseases

## Abstract

Replicative DNA polymerases, such as DNA polymerase α-primase, δ and ε, are multi-subunit complexes that are responsible for the bulk of nuclear DNA replication during the S phase. Over the last decade, extensive genome-wide association studies and expression profiling studies of the replicative DNA polymerase genes in human patients have revealed a link between the replicative DNA polymerase genes and various human diseases and disorders including cancer, intellectual disability, microcephalic primordial dwarfism and immunodeficiency. These studies suggest the importance of dissecting the mechanisms involved in the functioning of replicative DNA polymerases in understanding and treating a range of human diseases. Previous studies in *Drosophila* have established this organism as a useful model to understand a variety of human diseases. Here, we review the studies on *Drosophila* that explored the link between DNA polymerases and human disease. First, we summarize the recent studies linking replicative DNA polymerases to various human diseases and disorders. We then review studies on replicative DNA polymerases in *Drosophila*. Finally, we suggest the possible use of *Drosophila* models to study human diseases and disorders associated with replicative DNA polymerases.

## 1. Introduction

Eukaryotic DNA replication is an essential cellular event that is initiated by formation of the prereplication complex (preRC) [1,2,3]. The preRC is composed of the origin recognition complex (ORC), cell division cycle 6 (cdc6), chromatin licensing and DNA replication factor 1 (Cdt1) and the minichromosome maintenance (MCM) complex and is activated by a series of phosphorylation events that allow loading of another set of proteins to form the preinitiation complex (preIC). Subsequent phosphorylation events trigger loading of elongation proteins followed by the initiation of replication and synthesis of both the leading and the lagging strands. Replicative DNA polymerases, DNA polymerase α-primase, DNA polymerase δ and DNA polymerase ε, play major roles in the initiation and elongation processes of nuclear DNA replication [4,5,6]. DNA polymerase α-primase is mostly responsible for the synthesis of short RNA–DNA primers that initiate DNA synthesis. On the lagging strand, DNA polymerase α-primase repeatedly synthesizes short RNA–DNA primers and is then replaced by DNA polymerase δ to produce so-called Okazaki fragments. On the leading strand, after the synthesis of short RNA–DNA primers, DNA polymerase α-primase is replaced by DNA polymerase ε to carry out the leading strand DNA synthesis, although DNA polymerase δ may also be used in the leading strand synthesis under some circumstances. Unlike DNA polymerase α-primase, both DNA polymerases δ and ε possess a proofreading 3ʹ–5ʹ exonuclease activity that helps to ensure a high fidelity of DNA replication [4,5,6].

Extensive genome-wide association studies (EGWS) and expression profiling of the replicative DNA polymerase genes in human patients have revealed a link between the replicative DNA polymerase genes and various human diseases and disorders. In this review, we summarize the recent studies linking replicative DNA polymerases to human diseases and disorders and discuss the possible use of *Drosophila* models to study them.

## 2. Association of the Replicative DNA Polymerase Genes with Human Diseases and Disorders

### 2.1. Human DNA Polymerase α-Primase Genes

In all eukaryotes, DNA polymerase α-primase is composed of four subunits including two polymerase subunits and two primase subunits [6,7]. The human *POLA1* gene encodes the largest 166 kDa subunit that carries DNA polymerase activity (Table 1). The human *POLA2* gene encodes a 66 kDa subunit that plays a regulatory role for the DNA polymerase activity. The human *PRIM1* gene encodes a 50 kDa subunit carrying out the DNA primase activity and the human *PRIM2* gene encodes a 59 kDa subunit playing a regulatory role for the primase activity.

Although severe germline mutations in POLA1 very likely induce embryonic lethality, a combination of classical linkage analysis, targeted next-generation sequencing and whole-exome sequencing identified a unique intronic mutation in the POLA1 gene in several independent families presenting with X-linked reticulate pigmentary disorder (XLPDR, OMIM #301220), a rare syndrome characterized by skin hyperpigmentation, sterile multiorgan inflammation, recurrent infections and distinct facial features [8]. This hypomorphic mutation generates a new splice donor site in intron 13 of the POLA1 gene, which is localized at 76 nucleotides downstream of a cryptic splice acceptor sequence in the same intron, inducing mis-splicing of the pre-mRNA to create a new exon 13a [9]. In patients, the spliced POLA1 transcripts are decreased to 40–60%. On the other hand, probably due to its instability, the mis-spliced transcript containing the new exon 13a was expressed at very low levels [9]. In addition, there are no experimental data indicating that a truncated protein is expressed in XLPDR-derived cells [9]. Thus, XLPDR is probably caused by a nearly half-reduction of DNA polymerase activity, although it is still unknown whether this is due to a simple gene dose effect or the effect of some tissue-specific mis-splicing. More severe hypomorphic mutations in POLA1 that are distinct from the XLPDR-causing intronic mutation, have also been reported [8,10]. These mutations are known to be associated with van Esch–O’Driscoll syndrome (VEODS, OMIM#301030), manifesting growth retardation, microcephaly, hypogonadism, and immunological features similar to those seen in XLPDR [8,10]. A splice site mutation that was found in at least one of the families induced reduction of the POLA1 protein expression level much more than that observed in XLPDR [8,10]. In other VEODS families, the mutant POLA1 protein with decreased polymerase activity was expressed [8,10]. Therefore, this syndrome may reflect a state of a more profound or more widespread POLA1 defect that is accompanied by the growth retardation shared by all the affected VEODS patients. Interestingly, the clinical and immunological features observed in VEODS are similar to those observed with mutations in PRIM1, as well as in genes encoding other factors that function at the replication fork such as MCM4, MCM10, and GINS1 [8,9,10]. For further understanding of these disorders, development and utilization of animal models are expected.

In silico analysis of normal and malignant tissue expression data revealed that POLA1 levels are increased in colorectal cancer cells and tissues compared to non-transformed cells. ST1926, an atypical retinoid, showed a potent antitumor activity in colorectal cancer and also inhibited DNA polymerase α activity with decreased POLA1 protein expression levels [11]. Furthermore, mutations in POLA1 were found in the cells that became resistant to ST1926, suggesting POLA1 as a potential target in colorectal cancer treatment [11]. Other studies revealed that a combined treatment with extracts of plants, berberine and *Andrographis*, exhibits an enhanced anticancer activity in colorectal cancer [12]. Transcriptome analyses demonstrated that expression of DNA replication-related genes, including POLA1, is suppressed with treatment, and this is suggested to be responsible for the enhanced anticancer activity [12]. Longitudinal analysis of gene expression changes during cervical carcinogenesis identified several potential therapeutic target genes including POLA1 [13]. In addition, The Cancer Genome Atlas data analysis revealed several genes, including POLA1, that are enriched in high-risk hepatocellular carcinoma. The observed changes in POLA1 in multiple cancers suggest that a more detailed understanding of the roles that POLA1 plays in these cancers should aid in the development of effective therapies [14].

Cancer immunotherapy with adoptive transfer of tumour-infiltrating lymphocytes (TILs) appears to be effective for patients suffering metastatic melanoma. Whole-exome sequencing of DNA from the tumour cells identified the mutated POLA2 as one of the mutated antigens recognized by TILs [15]. It is thus suggested that recognition of the mutated POLA2 by TILs mediates the regression of cancer in these patients [15].

Microcephalic primordial dwarfism disorders (MPD) are characterized by intrauterine growth retardation, short stature and microcephaly [16]. Whole-genome and whole-exome sequencing of DNA from families with MPD identified biallelic mutations in PRIM1 [17]. These mutations extensively reduced the PRIM1 protein level in patient cells resulting in defects in replication fork formation and increased stalling of the DNA replication fork, reduced initiation of DNA replication at replication origins, and delayed cell cycle progression with a longer S phase [17]. These observed changes associated with mutations in the *PRIM1* gene could explain the severe growth failure of MPD.

Biliary atresia (BA) is a rare disease of infants in which the bile ducts outside and inside the liver are scarred and blocked, inducing damage in the liver. The damage induces scarring, loss of liver tissue and function and cirrhosis. Three types are defined for BA. Type I and Type II have atresia at the site of the common bile duct and at the site of the hepatic duct, respectively. The most commonly observed Type III has atresia at the site of the porta hepatis [18]. Whole-exome sequencing analysis of DNA from Type III BA patients identified mutations in the *PRIM2* and *MAP2K3* genes with high mutation rates [19]. However, the molecular basis of the effect of PRIM2 mutations on the pathogenesis of BA is not known yet. Association studies between single-nucleotide polymorphisms (SNPs) in mitotic phase-related genes and overall survivals of patients with non-small-cell lung cancer identified three independent SNPs in the *PRIM2*, *CHEK1* and *CDK6* genes, suggesting that these genetic variants are useful prognostic markers of the patients suffering non-small-cell lung cancer [20].

### 2.2. Human DNA Polymerase δ Genes

Human DNA polymerase δ is composed of four subunits, POLD1, POLD2, POLD3 and POLD4 (Table 1). The *POLD1* gene encodes the largest 125 kDa catalytic subunit carrying 5′–3′ DNA polymerase and 3′–5′ exonuclease activities. Mandibular hypoplasia, deafness, progeroid features and lipodystrophy (MDPL) syndrome (OMIM #615381) is a rare autosomal dominant systemic disorder characterized by progressive lipodystrophy with a lack of subcutaneous adipose tissue, mandibular hypoplasia, deafness and progeroid features. Several patients diagnosed with MDPL carried a deletion mutation of a serine residue at amino acid position 605 of POLD1 (S605del). The mutation was found to be heterozygous [21,22,23], indicating autosomal dominance. S605 is located in the polymerase active site, and a polymerase lacking this amino acid can bind DNA but is not able to catalyse polymerization. Biallelic mutations in POLD1 or POLD2 are also found in patients manifesting an autosomal recessive syndrome that combines replicative stress, neurodevelopmental abnormalities and immunodeficiency [24]. Homozygous missense variant D293N was found in the *POLD2* gene. This mutation is located in the phosphodiesterase domain of POLD2. Biallelic variants Q684H (located in the polymerase catalytic domain) and S939W (in the interdomain region) were found in the *POLD1* gene. Cultured cells from patients exhibited defects in cell cycle progression and replication-associated DNA lesions (such as fork stalling and fork collapse). Interestingly, these defects were rescued by overexpression of DNA polymerase δ. From detailed analyses, the mutations appear to affect the stability and subunit interactions within the DNA polymerase δ complex and/or its intrinsic DNA polymerase activity [24].

It is also reported that specific heterozygous germline mutations in the proofreading exonuclease-coding region and splicing sequences of POLE and POLD1 are associated with oligo-adenomatous polyposis, early-onset colorectal cancer and endometrial cancer [25]. It is estimated that genetic defects in POLD1 account for 0.2% of early-onset familial colorectal cancer [26]. Germline POLD1 variants associated with cancer include D316G/H, G321S, P327L, R409W, L474P, S478N and R507C in the exonuclease domain [23]. The S478N mutation in POLD1 appears to be responsible for the development of microsatellite-stable, chromosomal-unstable colorectal adenocarcinoma and/or oligopolyposis with high penetrance and dominant inheritance [27,28,29]. The L474P mutation in POLD1 is associated with hereditary nonpolyposis colorectal cancer. Although Lynch syndrome (also known as hereditary nonpolyposis) has been primarily associated with mutations in several MMR-related genes, mutations in POLD1, POLD2, POLD3 and POLD4 have more recently been identified as also being contributory [30]. These mutations in POLD1 may predispose to endometrial tumours and also breast and brain tumours [27,28,31]. In addition, alterations in expression or activity of DNA polymerase δ are related to senescence, aging and diabetes [23]. DNA polymerase δ plays a critical role at the junction of DNA replication with the repair and maintenance of genome integrity. Although further studies are necessary to understand the relationship between POLD1 mutations and the associated diseases, the genetic variants in POLD1 could be useful prognostic biomarkers of patients with various associated diseases.

### 2.3. Human DNA Polymerase ε Genes

Human DNA polymerase ε consists of four subunits, POLE (POLE1), POLE2, POLE3 and POLE4 (Table 1). The *POLE* gene, homologous to *Drosophila* PolE1, encodes the catalytic subunit of DNA polymerase ε carrying 5′–3′ DNA polymerase and 3′–5′ exonuclease activities. Facial dysmorphism, immunodeficiency, livedo and short stature (FILS) syndrome (OMIM #615139) is characterized by mild facial dysmorphism including high forehead and malar hypoplasia, livedo on the skin usually since birth, immunodeficiency that results in recurrent infections and short stature [32]. A homozygous single base substitution in the *POLE* gene induced alternative splicing in the conserved region of intron 34 that resulted in the reduction of POLE expression likely to cause FILS syndrome. In fact, the G1-to-S phase progression in the activated T lymphocytes from the patients was impaired as expected.

Intrauterine growth retardation, metaphyseal dysplasia, adrenal hypoplasia congenita, genital anomalies and immunodeficiency (IMAGEI) syndrome (OMIM#618336) is an autosomal recessive disorder characterized by intrauterine growth retardation, metaphyseal dysplasia, adrenal hypoplasia congenita, genital anomalies and immunodeficiency. Patients manifest distinctive facial features and variable immunodeficiency, including defects of lymphocytes. Compound heterozygosity for mutations in the *POLE* gene was found in IMAGEI syndrome patients, such as the same intronic variant as part of a common haplotype combined with different loss-of-function mutations [33]. Analysis of fibroblasts from patients revealed cellular deficiency of POLE and delay in S phase progression.

In some families with susceptibility to colorectal cancer (OMIM #615083), a heterozygous missense mutation in a highly conserved residue (L424V) in the proofreading exonuclease domain of POLE was identified [27]. The mutation showed autosomal dominant inheritance of a high-penetrance predisposition to the development of colorectal adenomas and carcinomas, with a variable tendency to develop multiple and large tumours. In addition, several somatic POLE mutations in the exonuclease domain were also identified in colorectal cancers from a large database [27]. Notably, all of these tumours carried additional somatic mutations, most commonly in the *APC* gene. It is suggested that the mechanism of tumorigenesis for some POLE-mutated tumours is probably due to a decrease in proofreading activity which causes an increased mutation rate. However, in some cases, the mechanism is not clear, and an additional defect is likely to be involved as the mutation rate observed for the cancer-related variant is higher than that associated with an exonuclease null mutant (reviewed in [34] and the references therein).

## 3. Studies on Replicative DNA Polymerases and Their Genes in *Drosophila*

As shown in Table 1, *Drosophila* has close orthologues of each of the human replicative polymerases.

### 3.1. Drosophila DNA Polymerase α-Primase

The requirement to replicate the entire genome during the very short early cycles in *Drosophila* embryos necessitates the rapid availability of large amounts of replication factors, including DNA polymerases. This led to the DNA polymerase α from *Drosophila* being one of the earliest eukaryotic replicative polymerases to be purified and characterised biochemically [35,36,37,38,39,40,41]. These early preparations of the enzyme suffered considerable proteolysis, but the adjustment of purification conditions allowed purification of an intact enzyme [42,43,44,45,46] which was observed to contain four subunits with sizes of 182 kDa (POLA1), 73 kDa (POLA2), 60 kDa (PRIM1) and 50 kDa (PRIM2) (Table 1). This was subsequently seen to be very similar to the subunit arrangements in DNA polymerase alpha enzymes from other species. Early characterisation with a purified protein showed that the polymerase activity was associated with POLA1, and primase activity—with Prim1 and Prim2 [38,39,40,41].

The capability to purify large amounts of the protein early on allowed extensive characterisation to be carried out on the enzyme in terms of its interactions with templates, processivity and fidelity [47,48,49,50,51,52,53,54,55,56,57]. Characterisation of subcomplexes of the enzyme also suggested that a subcomplex containing the POLA1 and POLA2 subunits of the enzyme might contain a cryptic exonuclease activity, although this has yet to be confirmed with purified proteins [58,59].

Cloning later confirmed the subunit sizes [60,61,62,63,64]. It showed that the subunits possessed significant homology to the equivalent human proteins: POLA1—40%, POLA2—30%, PRIM1—42%, PRIM2—33% (reviewed [6]). It also allowed the demonstration that all of the subunits were needed for embryonic viability [65,66]. In addition, specific depletion of POLA1, POLA2 and PRIM1 from eye discs resulted in perturbations in eye morphology [65,66,67]. Selection of mutants of POLA1 that were resistant to aphidicolin in cell culture also allowed characterisation of cellular strategies for evading inhibition by this reagent [68]. Analysis of the cloned primase subunits demonstrated that the active site of the primase was on PRIM2, but that PRIM1 seemed to help in maintaining the stability of the primase activity [63,64]. The precise role of POLA2 still remains unclear. For DNA polymerase α complexes from other species, it has been suggested to play a scaffolding role between the polymerase and the primase subunits, and given the homology between POLA2 orthologues from different species, it is likely that this function would be conserved.

Characterisation of expression patterns of the polymerase subunits allowed analysis of the cellular factors involved in their expression, identifying the involvement of the transcription factor DREF (DNA replication-related element binding factor), later seen to be important for the expression of a variety of replication-related factors [69,70,71,72,73].

Several interactors have been identified for *Drosophila* DNA polymerase α using a variety of techniques. These include an importin α protein involved in the nuclear transport of proteins [74], Hsp90-associated protein Dpit47 [75], histone methyltransferase PrSET7 [76], AAF (alpha-associated factor) [77], which is now more commonly known as the telomere-related CST complex [78], ctf4 (Chromosome Transmission Fidelity 4) [79], which is associated with the replication fork protection complex, kinase activity [80] and sumo conjugation cofactor uba2 [81]. Interactions that have been identified for DNA polymerase alpha from other organisms such as the interaction with histones [82] and mismatch repair proteins [83] are assumed to be conserved; however, they have not been confirmed experimentally.

POLA2 has been reported to be phosphorylated in early embryos, but aside from this, little is known about the post-translational modifications of *Drosophila* DNA polymerase α subunits [84].

### 3.2. Drosophila DNA Polymerase δ

This was purified later than DNA polymerase α due to the smaller amounts of the protein which are found in embryos. Early preparations of the pure enzyme had one or two polypeptides [85,86,87], while more recently, three subunits have been detected [88]. This is similar to the yeast DNA polymerase δ, which also has three subunits but differs from vertebrate polymerases which have four subunits. The homology of the two largest subunits to the human equivalents is quite high, but the third subunit has a lower similarity, perhaps reflecting the possible difference in subunit structure (POLD1—58%, POLD2 (POL31)—43%, POLD3 (POL32)—22%) (Table 1) [6].

As for other polymerases, polymerase and exonuclease activities are found in the largest subunit (POLD1). This subunit was cloned [89] and shown to be essential in mutants and by depletion [90,91,92]. POLD2 has not been characterised in *Drosophila*, but by analogy to other species is thought to be essential, maybe serving a scaffold function. POLD3 has been shown to be essential for replication [88,92], and has been suggested to be needed for the transport of the complex into the nucleus [88]. Loss of POLD3 also seems to affect DNA repair [93,94]. However, as for other species, this subunit also seems to form part of the repair DNA polymerase ζ, which makes it hard to directly assign any effects to its role as part of the δ complex.

In early studies, the purified enzyme was shown to be PCNA-dependent and more processive than DNA polymerase α [85,86,87]. Some substrate interactions have been characterised [95]; however, overall, very little biochemical characterisation has been carried out directly on the *Drosophila* enzyme. Because of this, its catalytic properties, interactions and PTMs are largely inferred from studies on delta enzymes from other species.

### 3.3. Drosophila DNA Polymerase ε

This was also originally purified from embryos as a one-subunit enzyme or a two-subunit enzyme [96,97] containing only the two largest subunits. The sequences of these subunits show high homology with their human orthologues [6,98] (POLE1—55%, POLE2—42%) (Table 1). The other two subunits which are found in the human enzyme have not formally been shown to be a part of the enzyme; however, the high homology between the *Drosophila* and human orthologues (POLE3—47%, POLE4—38%) makes it likely they will also be present in this enzyme (Table 1). Although their role in replication has not been studied, POLE3 and POLE4 have been seen to be involved in the CHRAC complex in *Drosophila* [99] and are also known as Chrac-14 (POLE3) and Mes4 (POLE4).

As for the other replicative polymerases, both polymerase and exonuclease activities are found in the N-terminal region of the large subunit. This subunit is essential for DNA replication in embryos and for endoreduplication in the salivary gland. Depletion in eye discs produces small eyes, and this effect, unlike those in embryos and salivary glands, can be partially recovered by expressing the C-terminus of the protein [100,101]. This region contains no apparent catalytic activity (although does contain inactivated polymerase and exonuclease regions), but does contain a number of protein interaction sites. Similar results have also been obtained for DNA polymerase ε from *Schizosaccharomyces pombe* [102]. POLE2 has also been shown to be essential for embryos and has a role in endoreduplication [103].

As for DNA polymerase δ, only limited characterisation has been carried out on purified *Drosophila* DNA polymerase ε, and all this with the one- or two-subunit version of the enzyme. The purified enzyme was shown to be highly processive and some information about its substrate affinities was determined [96,97]. Genetic interaction studies, however, have suggested an interaction between the polymerase and a number of other proteins. The large subunit has been seen to interact with replication proteins RFC, DNA primase, DNA polη, Mcm10 and Psf2 and with the chromatin remodelling enzyme Iswi [100]. In addition, POLE2 has been observed to interact with ORC2 [103].

## 4. Possible Application of *Drosophila* Models to Study Human Diseases Associated with Replicative DNA Polymerase Genes

Although the utility of the *Drosophila* model to study human disease is well-established, it is at present underused to develop models for diseases specifically associated with replicative DNA polymerase genes [3]. Mutations in replicative DNA polymerase genes are often associated with rare intractable diseases and disorders such as XLPDR, VEODS, MPD, BA, MDPL syndrome, immunodeficiency with replicative stress, FILS syndrome and IMAGEI syndrome (Table 1). Some of these disorders are associated with decreased expression levels or loss of large parts of the protein, but some are associated with point mutations, and many of the amino acids mutated in disease are conserved in *Drosophila*.

In comparison to other model systems, however, *Drosophila* has a number of advantages. *Drosophila* is easy to maintain and manipulate in the laboratory. *Drosophila* has a relatively short generation time, and production of a large number of offspring guarantees statistical significance of the obtained experimental data (Table 2) [104,105,106]. Mutants, overexpression and RNAi lines for many of the *Drosophila* replicative DNA polymerase genes are available from various stock centres. In addition, in many countries, experimentation with *Drosophila* is often encouraged as an alternative to working with mammals. Thus, *Drosophila* could provide useful models to elucidate pathogenesis of these human diseases and disorders, develop novel diagnostic markers and therapeutic targets and screen for candidate therapeutic substances.

Human diseases and disorders associated with replicative DNA polymerase gene mutations often show complex symptoms affecting many body systems. In terms of detailed phenotypic analysis, the physical structures affected by these disorders are all found in *Drosophila*, and in addition, many health and behavioural analysis techniques such as viability, fertility and lifespan assays, locomotive assays with larvae and adults, activity monitoring to examine the circadian rhythm and learning assays are already well-established. Therefore, as shown in Figure 1, these assays with suitable *Drosophila* models would be useful for studying the effects of patient-derived mutations on development and survival.

Genetic screens are useful to search for novel diagnostic markers and therapeutic targets for diseases and disorders. There is ample precedent in using such screens in *Drosophila*, including some examples with polymerase genes: specific depletion of POLA1, POLA2 and PRIM1 from eye discs resulted in perturbations in eye morphology [64,65,66]. Wing disc-specific knockdown of PolA1 induces an atrophied wing phenotype [76]. Using these fly lines, dPrSet7, a gene encoding histone H4 lysine 20-specific methyltransferase (PrSET7) was successfully identified [76]. These flies with eye or wing phenotypes could also be useful to identify other genetic interactants with DNA polymerase α genes, which could in turn be useful to identify diagnostic markers and therapeutic targets for XLPDR, VEODS, MPD and BA with which DNA polymerase α genes are associated. Eye disc-specific knockdown of PolE1 induces a small eye phenotype (unpublished data). Therefore, this fly line may be useful to identify genetic interactants with a DNA polymerase ε gene and further characterize the pathogenesis of FILS syndrome and IMAGEI syndrome.

Mutations in various replicative DNA polymerase genes are often associated with colorectal and endometrial cancers (Table 1). In *Drosophila*, cancer models have been established and used to study various aspects of cancer research [107]. *Drosophila* tumour suppressor lgl cooperates with scribble and discs sufficiently large to maintain apicobasal polarity of epithelial cells [106]. Their mutations induce disorganization of epithelial tissue architecture followed by formation of multi-layered metastatic tumours. Genetic mosaic analysis in *Drosophila* allowed recapitulation of various aspects of human cancers, such as clonal evolution, cancer cachexia, tumour microenvironment and anticancer drug resistance [107]. In fact, the mosaic analysis contributed to elucidation of hyperplastic tumour suppressor genes in the Hippo signalling pathway, a tumour suppressor pathway that is highly conserved between *Drosophila* and humans [107]. Following a similar methodology, these *Drosophila* cancer models could be used to dissect the possible involvement of replicative DNA polymerase genes in the generation and maintenance of cancer. Recently identified mutations in various replicative DNA polymerase genes, which are often associated with colorectal and endometrial cancers (Table 1), offer good candidates for the use of such an approach. The ease with which gene editing can be accomplished in *Drosophila* allows the introduction of patient-derived point mutations into genomic copies of DNA polymerase genes. This would allow monitoring of the effects of these mutations on the whole organism during development and also provide a useful system to test potential therapeutic substances.

## 5. Perspectives

The advent of techniques such as Crispr–Cas9 has expanded the number of model organisms that can be genetically manipulated to allow the study of the effects of disease-related mutations in a whole-organism setting. However, the rapid life cycle and relatively modest expense of *Drosophila* maintenance together with the ability to handle large population sizes still gives *Drosophila* advantages over other multicellular organisms. *Drosophila* has already made a significant contribution to the molecular understanding of the way that cells function; exploitation of its experimental advantages should also lead to significant contributions to the understanding and treatment of disease-related mutations in human polymerases.

## Figures and Tables

**Figure 1 ijms-24-08078-f001:**
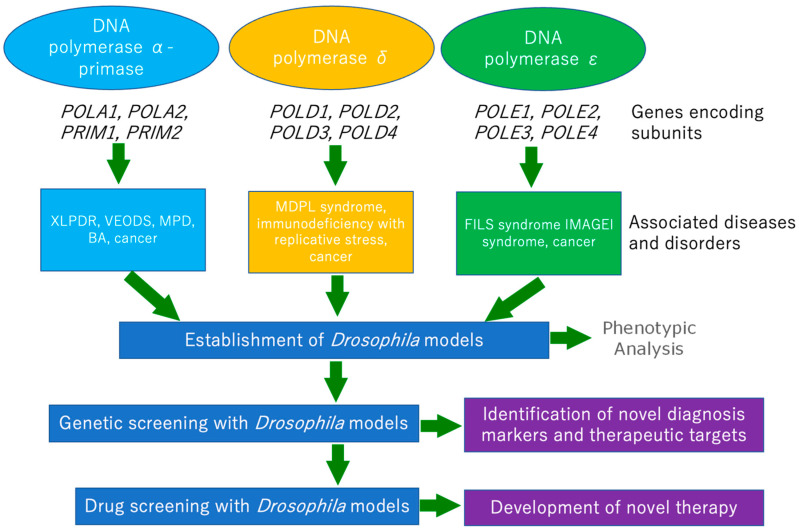
Schematic outlining the possible use of *Drosophila* models to study human replicative polymerase disorders. XLPDR—X-linked reticulate pigmentary disorder, VEODS—an Esch–O’Driscoll syndrome, BA—Biliary atresia, MPD—Microcephalic primordial dwarfism disorders, MDPL—Mandibular hypoplasia, deafness, progeroid features, and lipodystrophy syndrome, FILS—Facial dysmorphism, immunodeficiency, livedo and short stature syndrome, IMAGE—Intrauterine growth retardation, metaphyseal dysplasia, adrenal hypoplasia congenita, genital anomalies, and immunodeficiency syndrome.

**Table 1 ijms-24-08078-t001:** Diseases associated with mutations in human replicative DNA polymerase subunits and their associated *Drosophila* orthologues [6].

	Replication Function	Subunits and Encoding Gene (Gene Symbol)	Subunit Mass in kDa (Human)	UniProt Protein Accession	Human Orthologue	Identity (%)	Disease Related to Mutation (Mutation Type) [8,9,10,11,12,13,14,15,16,17,18,19,20,21,22,23,24,25,26,27,28,29,30,31,32,33]
DNA polymerase α-primase	Synthesis of the initiation primer and Okazaki fragment primers	POLA1 (*PolA1*)	182 (180)	P26019	POLA1	40	XLPDR (intronic), VEODS (intronic), colorectal cancer
		POLA2 (*PolA2*)	73 (70)	Q9VB62	POLA2	30	Melanoma
		PRIM1 (*Prim1*)	60 (58)	Q24317	PRIM1	42	MPD (intronic)
		PRIM2 (*Prim2*)	50 (49)	Q9VPH2	PRIM2	33	BA (intronic), lung cancer
DNA polymerase δ	Bulk lagging strand synthesis Initiation and termination of the leading strand synthesis Proofreading some regions of the DNA polymerase ε synthesis	POLD1 (*PolD1*)	125 (125)	P54358	POLD1	58	MDPL syndrome (S605del), immunodeficiency with replicative stress (Q684H, S939W), colorectal and endometrial cancer (D316G/H, G321S, P327L, R409W, L474P, S478N, R507C)
		POLD2 (*PolD2*)	48(68)	Q9W088	POLD2	43	Immunodeficiency with replicative stress (D293N), colorectal and endometrial cancer
		POLD3 (*PolD3*)	47 (50)	Q9Y118	POLD3	22	Colorectal and endometrial cancer
		–	– (12)	–	POLD4		
DNA polymerase ε	Bulk leading strand synthesis	POLE1 (*PolE1*)	257 (261)	Q9VCN1	POLE	55	FILS syndrome (intronic), IMAGEI syndrome (intronic), colorectal cancer (L424V)
		POLE2 (*PolE2*)	59 (59)	Q9VRQ7	POLE2	42	Colorectal cancer
		POLE3 (*PolE3*)	14 (14)	Q9V444	POLE3	47	
		POLE4 (*PolE4*)	17 (12)	Q9W256	POLE4	38	

**Table 2 ijms-24-08078-t002:** Comparison of *Drosophila* with other models which allow whole-organism analysis [104,105,106].

Features	*Drosophila*	Mouse	Zebra fish	Nematode	Yeast
Genome size (bp)	0.14 × 10^9^	2.8 × 10^9^	1.4 × 10^9^	0.10 × 10^9^	0.012 × 10^9^
Protein-coding genes shared with humans	58%	88%	70%	60–80%	30%
Human disease genes conserved	85%	99%	84%	65%	20%
Generation time	10 days	9–11 weeks	8–17 weeks	4 days	80 min
Lifespan	70–90 days	1.3–3 years	3–5 years	18–20 days	NA
Production of offspring/female	100 eggs/day	10/litter	200–300 eggs/week	300–350 eggs	NA
Ethical restriction	+	+++	+++	+	NA
Large-scale genetic screening	+++	+	++	+++	++++
Large-scale drug screening	+++	+	++	+++	+++

More + indicates that the parameter considered is more relevant for that organism. NA indicates that the parameter is not applicable to that organism. N/A.

## Data Availability

Not applicable.

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
