# Peer review of "Association of Mutations in Replicative DNA Polymerase Genes with Human Disease: Possible Application of Drosophila Models for Studies"

_ijms, 2023, doi:10.3390/ijms24098078_

Round 1

Reviewer 1 Report

DNA polymerase gene alterations in humans cause various diseases. The current manuscript advocates for Drosophila as a model to get insights into underlying mechanisms. In the first part, the authors describe the connection between polymerase alterations and human diseases. That is done superficially and incompletely. We understand that the literature on the subject is immersing, but several seminal papers and findings are still not discussed, and many reviews on the subject are missed (e.g.,  PMID: 32883112; PMID: 30818169PMID: 28687338PMID: 35326618:PMID: 35780178). In the second part, the authors focus on the possible use of Drosophila as a model for studying the effects of polymerase mutations. The reader sees that the same polymerases are present in flies (not surprising), but, unfortunately, more needs to be done with polymerase mutations experimentally besides several tissue-specific knockdowns. The review needs to persuade the reader that Drosophila is the best or at least a good model. Comparisons in Table 2 are made only with the mice model, neglecting other objects, like yeast, C. elegans, plants, and cancer cell lines. Additional information regarding the availability of genetic manipulation techniques and the power of analysis of phenotypes should be provided in this Table. The manuscript does not have illustrations that might help grab the reader's attention.

Minor comments:

Title: Maybe better to write “association with the status, or alteration, or mutations of genes”, not polymerase genes.

Abstract: The sentence "...the importance of the polymerase genes for therapy and diagnosis" looks generic; some specifics should be added.

Page 1. It should be "proofreading" instead of "proof reading".

Table 1. The title should be edited because the Table contains information about human diseases. The author should also write if they consider only germline variants. Do the authors use the correct Drosophila nomenclature for genes and proteins? We suggest avoiding bulk citations in the description of the last column. Instead, every entry in this column should have a supporting citation. The reader interested in a particular protein should be able to find references without browsing the whole list. 

Page 4. The author should better explain how POLA1 could be used for cancer therapy. 

The next paragraph has an allusion to POLB that is not discussed elsewhere. 

What are “ replication associated DNA lesions…”?

Page 5. MMR gene mutations were identified as cancer causes much earlier than polymerase mutations. 

The mechanism of increased mutation rates in POLE tumors is not well understood; it seems that not only proofreading matters PMID: 30670691, PMID: 35822874

Page 6. Mentioning “cryptic exonuclease activity” described more than twenty years ago and never confirmed could be omitted.

 Page 7. Is the sharing of subunits of pol δ and pol ζ confirmed in Drosophila? The authors write, "also seems".

Ref 34 is marked as ref 1.

Author Response

Thankyou to the referee for their helpful comments. Below we have outlined our responses to these. 

We understand that the literature on the subject is immersing, but several seminal papers and findings are still not discussed, and many reviews on the subject are missed 

We are not sure what seminal papers have been missed. We would be happy to include these if the referee could point us towards the omitted papers. 

We have also included the most recent clinical review on cancer prone polymerases ( together with ‘and references therein’). The main aim of  the review is to show the breadth of diseases that mutations in the polymerases are involved in and we do not want to put much emphasis on any one disease in the references. 

In the second part, the authors focus on the possible use of Drosophila as a model for studying the effects of polymerase mutations. The reader sees that the same polymerases are present in flies (not surprising), but, unfortunately, more needs to be done with polymerase mutations experimentally besides several tissue-specific knockdowns. The review needs to persuade the reader that Drosophila is the best or at least a good model. 

 We have  altered section 4 in several places and included a perspectives section at the end.We have also included a new figure ( Figure 1) that lays out a flow for possible experimentation with Drosophila. Hopefully this may now make our case a little clearer. 

Comparisons in Table 2 are made only with the mice model, neglecting other objects, like yeast, C. elegans, plants, and cancer cell lines. 

We have now included yeast,c.elegans and zebra fish in table 2 so that a more complete comparison can be made. 

Additional information regarding the availability of genetic manipulation techniques and the power of analysis of phenotypes should be provided in this Table. 

Additional data has now been included in this table to cover this suggestion.

The manuscript does not have illustrations that might help grab the reader's attention.

We have added an additional figure ( see above ) which we hope will help with this.

Minor comments:

Title: Maybe better to write “association with the status, or alteration, or mutations of genes”, not polymerase genes.

We have changed this to “association of mutations in replicative polymerase genes ……….  “

Abstract: The sentence "...the importance of the polymerase genes for therapy and diagnosis" looks generic; some specifics should be added.

We have changed this to  ‘These studies suggest the importance of dissecting the mechanisms involved in the functioning of replicative DNA polymerases in understanding and treating a range of human diseases.’

Page 1. It should be "proofreading" instead of "proof reading".

This has been changed

Table 1. The title should be edited because the Table contains information about human diseases. 

This has been changed to ‘Diseases associated with mutations in human replicative DNA polymerase subunits and their associated Drosophila orthologues.’ 

The author should also write if they consider only germline variants. 

We have tried to make clear in the text where we are dealing with germline and where somatic mutations.

Do the authors use the correct Drosophila nomenclature for genes and proteins?  

We use the nomenclature as set out in reference 6 in the paper which we believe to be the correct one.  

We suggest avoiding bulk citations in the description of the last column. Instead, every entry in this column should have a supporting citation. The reader interested in a particular protein should be able to find references without browsing the whole list. 

These are already referenced individually next to the discussion of the disease in the text. This should enable the reader to find the reference quickly by referring to the relevant section rather than looking through the whole list.  

Page 4. The author should better explain how POLA1 could be used for cancer therapy. 

We have added ‘The observed changes in POLA1 in multiple cancers suggest that a more detailed understanding of the roles that POLA1 plays in these cancers should aid in the development of effective therapies.’ 

The next paragraph has an allusion to POLB that is not discussed elsewhere. 

We have changed Change polB to POLA2

What are “ replication associated DNA lesions…”?

We have added ‘such as fork stalling and fork collapse’ 

Page 5. MMR gene mutations were identified as cancer causes much earlier than polymerase mutations. 

We have added ‘ Although Lynch syndrome (also known as …) has been primarily associated with mutations in several mmr related genes, mutations in POL… have more recently been identified as also being contributory’ 

The mechanism of increased mutation rates in POLE tumors is not well understood; it seems that not only proofreading matters PMID: 30670691, PMID: 35822874

We have added ‘It is suggested that the mechanism of tumorigenesis for some POLE-mutated tumors is probably due to decreased fidelity of proofreading activity to induce an increased mutation rate.  However in some cases the mechanism is not clear and  an additional defect may is likely to be involved as the mutation rate observed for the cancer related variant is higher than that associated with an exonuclease null mutant’ and added in additional references that address this. 

Page 6. Mentioning “cryptic exonuclease activity” described more than twenty years ago and never confirmed could be omitted.

Although it is not necessary to leave this in, we don’t feel that it detracts from the paper. It was an observation that was made,  and to our knowledge has not been proved or disprove using cloned subunits.

 Page 7. Is the sharing of subunits of pol δ and pol ζ confirmed in Drosophila? The authors write, "also seems".

Yes according to reference 6 

Ref 34 is marked as ref 1.

Thankyou for spotting that this has been changed.

Reviewer 2 Report

Thank you for sending me this interesting manuscript by Masamitsu Yamaguchi and Sue Cotterill.  Overall, I judge this review as a good contribution to the understanding of the polymerase’s genes for therapy and diagnosis . The authors describe studies in Drosophila that explore the link between polymerases and human diseases with emphasis on DNA polymerase alpha, delta and epsilon and those mutations associated to rare diseases.

Authors suggest the possible use of Drosophila model for studying human diseases and disorders associated with the replicative DNA polymerases.

Overall, this manuscript is suitable for publication after addressing the points enlisted.

There are minor issues that are listed in order, as follows:

1) Page 3 : “In other VEDOS families, the mutant POLA1 protein with decreased polymerase activity is expressed [8,10]. Therefore, this…” change VEDOS to VEODS.

2) Page 5: “The L474P mutation n POLD1 is associated with hereditary non-polyposis colorectal cancer” change to “The L474P mutation in POLD1 is associated with hereditary non-polyposis colorectal cancer

3) I would like to see more figures and/or protein homology models indicating some of the mutations mentioned in this review.

Author Response

  1. Page 3 : “In other VEDOS families, the mutant POLA1 protein with decreased polymerase activity is expressed [8,10]. Therefore, this…” change VEDOS to VEODS.

This has been altered.  

2) Page 5: “The L474P mutation n POLD1 is associated with hereditary non-polyposis colorectal cancer” change to “The L474P mutation in POLD1 is associated with hereditary non-polyposis colorectal cancer”

This has been altered.

3) I would like to see more figures and/or protein homology models indicating some of the mutations mentioned in this review.

We have searched and seen that many of the amino acids involved in point mutations in human diseases are conserved in Drosophila and have added a sentence at the end of the first paragraph of section 4 stating this:

“Some of these disorders are associated with decreased expression levels or loss of large parts of the protein, but some are associated with point mutations, and many of the amino acids mutated in disease are conserved in Drosophila”. 

Reviewer 3 Report

This review show a summary of the works about human DNA polymerases and the link with disease such  as cancer and the useful tool that is Drosophilla to understand this relation. This work achieve a wide searching of the literature and review the topics properly. However, some details must be improved:

1. Table I appears without any relation with the text. It must be positioned in the text according the reference in the text in section 3.

2. The importance to compare both DNA polymerase machineries (human and drosophila) are a key topic of this review. For that reason, I suggest a comparative table between both machineries or the addition of a Table showing the subunits of the human DNA polymerase.  The Table related to Drosophila DNA polymerase is not enough if one of the goals of this review is validate the use of drosophila as a model to study the link between defects in human DNA polymerase subunits and diseases such as cancer.

3. I think that it is useful to add a scheme/model to the text where authors represent the association of human DNA polymerase genes and diseases.

4. It is necessary to add a final paragraph of "projections" or "conclusions".

Author Response

  1. Table I appears without any relation with the text. It must be positioned in the text according the reference in the text in section 3.

We have referenced Table 1 in section 3 of the text as suggested. We have also maintained the referrals to this table in section 2 as it is also relevant in this position.

2. The importance to compare both DNA polymerase machineries (human and drosophila) are a key topic of this review. For that reason, I suggest a comparative table between both machineries or the addition of a Table showing the subunits of the human DNA polymerase.  The Table related to Drosophila DNA polymerase is not enough if one of the goals of this review is validate the use of drosophila as a model to study the link between defects in human DNA polymerase subunits and diseases such as cancer.

Table 1 already contains information about the subunits of the Human DNA polymerase  and the degree of homology between the human and Drosophila polymerases. For further clarification we have now included human  POLD4 in the table, and also the molecular  weights of the human polymerase subunits.This now shows that the subunits are of good homology and comparable size which suggests that the Drosophila subunits should be a good model for the Human proteins. 

3. I think that it is useful to add a scheme/model to the text where authors represent the association of human DNA polymerase genes and diseases.

We have now added a schematic to the text ( Figure 1) depicting the relationship between the polymerase subunits and human disease as well as suggesting possible routes for further study.

 4. It is necessary to add a final paragraph of "projections" or "conclusions".

We have added a perspectives paragraph to the end of section 4. 

Round 2

Reviewer 1 Report

The manuscript is improved. 

One important thing to edit: on page 5 the authors incorrectly wrote about the link of proofreading and fidelity. 

Defective proofreading lowers fidelity because polymerization errors are not corrected, but the authors wrote about the fidelity of proofreading., which is in the context is incorrect.

Author Response

Thankyou for spotting this. this has been changed to  ‘…due to a decrease in proofreading activity which causes an increased mutation rate’.